# Characterising the Features of 381 Clinical Studies Evaluating Transcutaneous Electrical Nerve Stimulation (TENS) for Pain Relief: A Secondary Analysis of the Meta-TENS Study to Improve Future Research

**DOI:** 10.3390/medicina58060803

**Published:** 2022-06-14

**Authors:** Mark I. Johnson, Carole A. Paley, Priscilla G. Wittkopf, Matthew R. Mulvey, Gareth Jones

**Affiliations:** 1Centre for Pain Research, School of Health, Leeds Beckett University, Leeds LS1 3HE, UK; c.paley@leedsbeckett.ac.uk (C.A.P.); p.wittkopft@leedsbeckett.ac.uk (P.G.W.); g.j.jones@leedsbeckett.ac.uk (G.J.); 2Research & Development Department, Airedale NHS Foundation Trust, Steeton, Keighley BD20 6TD, UK; 3Academic Unit of Primary and Palliative Care, Leeds Institute of Health Sciences, University of Leeds, Leeds LS2 9NL, UK; m.r.mulvey@leeds.ac.uk

**Keywords:** transcutaneous electrical nerve stimulation (TENS), pain, pain management, therapeutic neuromodulation, secondary analysis

## Abstract

*Background and Objectives*: Characterising the features of methodologies, clinical attributes and intervention protocols, of studies is valuable to advise directions for research and practice. This article reports the findings of a secondary analysis of the features from studies screened as part of a large systematic review of TENS (the meta-TENS study). *Materials and Methods*: A descriptive analysis was performed on information associated with methodology, sample populations and intervention protocols from 381 randomised controlled trials (24,532 participants) evaluating TENS delivered at a strong comfortable intensity at the painful site in adults with pain, irrespective of diagnosis. *Results*: Studies were conducted in 43 countries commonly using parallel group design (*n* = 334) and one comparator group (*n* = 231). Mean ± standard deviation (SD) study sample size (64.05 ± 58.29 participants) and TENS group size (27.67 ± 21.90 participants) were small, with only 13 of 381 studies having 100 participants or more in the TENS group. Most TENS interventions were ‘high frequency’ (>10 pps, *n* = 276) and using 100 Hz (109/353 reports that stated a pulse frequency value). Of 476 comparator groups, 54.2% were active treatments (i.e., analgesic medication(s), exercise, manual therapies and electrophysical agents). Of 202 placebo comparator groups, 155 used a TENS device that did not deliver currents. At least 216 of 383 study groups were able to access other treatments whilst receiving TENS. Only 136 out of 381 reports included a statement about adverse events. *Conclusions*: Clinical studies on TENS are dominated by small parallel group evaluations of high frequency TENS that are often contaminated by concurrent treatment(s). Study reports tended focus on physiological and clinical implications rather than the veracity of methodology and findings. Previously published criteria for designing and reporting TENS studies were neglected and this should be corrected in future research using insights gleaned from this analysis.

## 1. Introduction

Transcutaneous electrical nerve stimulation (TENS) is used to alleviate the intensity of pain and involves the delivery of pulsed electrical currents across the skin to stimulate peripheral nerves. Physiological research demonstrates that TENS reduces activity and excitability of central projection neurons reducing nociceptive input to the brain and modulating pain experience [1,2,3,4]. TENS has been used globally for symptomatic relief of pain since the 1970s and TENS equipment is available without prescription in many countries [5]. TENS treatment is usually self-administered as often as is needed with minimal risk of adverse effects or toxicity. TENS equipment and clinical support is inexpensive, and health economic analyses suggest TENS lowers costs for persistent pain [6], chronic low back pain [7,8] and knee osteoarthritis [9]. It is not possible to predict with certainty who is likely to respond to TENS, although a 30-min TENS treatment has been shown to forecast the likelihood of longer-term outcome in women with fibromyalgia [10].

The debate about the effectiveness of TENS is long-standing, despite a wealth of published research studies spanning over five decades [11]. The consequence is inconsistency in clinical recommendations about prescribing TENS in the U.K. National Health Service (NHS), or coverage by private healthcare insurance in the Unites States. Recently, we published a systematic review and meta-analysis of 381 studies that found moderate certainty evidence that strong non-painful TENS lowered pain intensity when compared with placebo (i.e., the Meta-TENS study [12]).

There have been no analyses of the features of clinical studies on TENS. An investigation of trends, strengths, weaknesses and gaps in study methodologies, including TENS intervention protocols would be valuable to inform directions for future research and clinical practice. The purpose of this article is to report a secondary analysis of the characteristics of studies that was not reported in our original systematic review and meta-analysis of TENS (the Meta-TENS study) [12].

## 2. Methods

The Meta-TENS study was registered on PROSPERO (CRD42019125054), published as a protocol in 2019 [13] and in the primary report of the findings in 2022 [12]. Readers are referred to these publications for methodological detail and findings of the Meta-TENS study. Ethical approval for the Meta-TENS study was given by Leeds Beckett University (Application Ref: 78097). Here we provide the methodology used in our secondary analysis of study characteristics beyond that provided in our primary report.

Electronic databases were searched from inception to 17 May 2020 for randomised controlled trials (RCTs) evaluating TENS at the site of pain versus placebo, no treatment, or other treatments in adults experiencing pain regardless of diagnosis. Two independent reviewers extracted a variety of information from study reports including:
Methodological characteristics of studies (e.g., overall risk of bias, study group study size, concurrent use of other treatments).Pain characteristics (e.g., duration (acute, chronic), medical diagnosis (pain condition), mechanistic descriptor (nociceptive, neuropathic), physiological system (musculoskeletal, visceral, somatosensory).Intervention characteristics (e.g., high-frequency TENS, low-frequency TENS, types of placebos, types of comparator treatments).

One reviewer (MIJ) performed descriptive analyses by plotting bar charts and histograms and reported summary measures as frequency counts, means or medians as appropriate.

## 3. Results

### 3.1. Observations: Searching

There has been a steady rise in the rate of publication of studies on the efficacy of TENS since the 1970s as reflected in a search of Pubmed.gov conducted on 31 March 2022 (Figure 1).

In our Meta-TENS study [12], a total of 381 studies with 383 distinct populations and 24,532 participants were eligible for inclusion [14,15,16,17,18,19,20,21,22,23,24,25,26,27,28,29,30,31,32,33,34,35,36,37,38,39,40,41,42,43,44,45,46,47,48,49,50,51,52,53,54,55,56,57,58,59,60,61,62,63,64,65,66,67,68,69,70,71,72,73,74,75,76,77,78,79,80,81,82,83,84,85,86,87,88,89,90,91,92,93,94,95,96,97,98,99,100,101,102,103,104,105,106,107,108,109,110,111,112,113,114,115,116,117,118,119,120,121,122,123,124,125,126,127,128,129,130,131,132,133,134,135,136,137,138,139,140,141,142,143,144,145,146,147,148,149,150,151,152,153,154,155,156,157,158,159,160,161,162,163,164,165,166,167,168,169,170,171,172,173,174,175,176,177,178,179,180,181,182,183,184,185,186,187,188,189,190,191,192,193,194,195,196,197,198,199,200,201,202,203,204,205,206,207,208,209,210,211,212,213,214,215,216,217,218,219,220,221,222,223,224,225,226,227,228,229,230,231,232,233,234,235,236,237,238,239,240,241,242,243,244,245,246,247,248,249,250,251,252,253,254,255,256,257,258,259,260,261,262,263,264,265,266,267,268,269,270,271,272,273,274,275,276,277,278,279,280,281,282,283,284,285,286,287,288,289,290,291,292,293,294,295,296,297,298,299,300,301,302,303,304,305,306,307,308,309,310,311,312,313,314,315,316,317,318,319,320,321,322,323,324,325,326,327,328,329,330,331,332,333,334,335,336,337,338,339,340,341,342,343,344,345,346,347,348,349,350,351,352,353,354,355,356,357,358,359,360,361,362,363,364,365,366,367,368,369,370,371,372,373,374,375,376,377,378,379,380,381,382,383,384,385,386,387,388,389,390,391,392,393,394] (see our primary report for details [12], including studies awaiting classification [395,396,397,398,399,400,401,402,403,404,405,406,407,408,409,410,411,412,413] and studies excluded at full text screening with reasons [414,415,416,417,418,419,420,421,422,423,424,425,426,427,428,429,430,431,432,433,434,435,436,437,438,439,440,441,442,443,444,445,446,447,448,449,450,451,452,453,454,455,456,457,458,459,460,461,462,463,464,465,466,467,468,469,470,471,472,473,474,475,476,477,478,479,480,481,482,483,484,485,486,487,488,489,490,491,492,493,494,495,496,497,498,499,500,501,502,503,504,505,506,507,508,509,510,511,512,513,514,515,516,517,518,519,520,521,522,523,524,525,526,527,528,529,530,531,532,533,534,535,536,537,538,539,540,541,542,543,544,545,546,547,548,549,550,551,552,553,554,555,556,557,558,559,560,561,562,563,564,565,566,567,568,569,570,571,572,573,574,575,576,577,578,579,580,581,582,583,584,585,586,587,588,589,590,591,592,593,594,595,596,597,598,599,600,601,602,603,604,605,606,607,608,609,610,611,612,613,614,615,616,617,618,619,620,621]). It was noteworthy that an additional 36 studies were found and met our inclusion criteria between the initial search during July 2019 and the updated search on 17 May 2020. This demonstrates the high publication rate of studies on TENS, although only one of these additional studies had a study group sample size of at least 50 participants.

### 3.2. Observations: The Screening Process

#### 3.2.1. Few Instances of Multiple Records (Secondary Reports)

There were five instances of multiple records (secondary reports) of one study:A primary report of 70 participants with follow-up data at 1 year by Cherian et al. [79]; secondary reports of data of the first 23 participants [622] and 36 participants (presumably including the first 23 patients) [623] at 3 months.A primary report by Chesterton et al. [80] of TENS added to usual care for tennis elbow; secondary report of an economic evaluation by Lewis et al. [624].A primary report by Escortell-Mayor et al. [123] of TENS versus manual therapy for neck pain; additional Spanish language version equivalent [625].A primary report by Oosterhof et al. [267] of short-term outcome of TENS for various chronic pains; secondary reports of predictors of TENS outcome [626], long-term outcomes [627] and physiological mechanisms [628].A primary report by Pietrosimone et al. [281] on TENS for knee osteoarthritis and a secondary report of outcomes associated with knee kinematics and kinetics [629].

#### 3.2.2. Few Instances of Multiple Samples within Study Reports

There were two instances of distinct samples in one study report:Chia et al. [81] performed independent evaluations of nulliparous and multiparous participants combined (*n* = 101) and nulliparous only (*n* = 20).Kayman-Kose et al. [184] performed distinct evaluations of caesarean section (*n* = 100) and vaginal delivery (*n* = 100).

In our systematic review, we managed this issue by categorising each report as having distinct sample populations within one study and analysed data from these samples separately, i.e., we identified 383 different samples from 381 studies.

#### 3.2.3. No Instances of Duplication of Participants within Study Reports

Double counting of participant samples contributes to unit of analysis issues in meta-analyses. We did not detect any instances of data collected in published ‘pilot’ studies being included in ‘full’ study samples, although unclear reporting hindered our ability to detect duplication of participant data with certainty. For example, Lin et al. published a study of TENS for shoulder pain [217] and chronic shoulder tendonitis [218] but inspection of the reports suggested protocols and data were different. Thus, we considered these to be distinct study populations.

#### 3.2.4. Few Instances of Inconsistencies of Extracted Data with Previous Meta-Analyses

There were few inconsistencies on study data extracted for our Meta-TENS study [12] and previously published meta-analyses that were included in an overview of systematic reviews [630]. There were occasional instances of double counting of groups in pooled data analyses in previous reviews [631,632,633,634] contributing to unit of analysis errors. We detected inclusion of data from the study by Bjersa and Andersson [56] that was area under the curve rather than 0–100 mm VAS in a meta-analysis by Zhou et al. [635]). These discrepancies did not affect conclusions of the previously published reviews.

### 3.3. Features of Excluded Studies

The main reasons that studies did not meet eligibility criteria for our Meta-TENS study is provided in Figure 2. Many reports stated that a study evaluated TENS, although the intervention violated our criteria for standard TENS or was applied to an inappropriate body site. It was not possible to isolate TENS effects from concurrent treatment in at least 17 studies.

### 3.4. Not Standard Electrical Characteristics

At least 90 studies used interventions that were not ‘standard TENS’ (i.e., type of stimulating device and accessories and electrical characteristics of stimulation): Codetron (acupuncture-like stimulation); auto-targeted neurostimulation; frequency rhythmic electrical modulation; H-wave therapy; high-voltage pulsed direct current; interferential current therapy; InterX (non-invasive interactive neurostimulation); microcurrent electrical stimulation; neuromuscular electrical stimulation; supraorbital transcutaneous stimulation; transcutaneous electric acupoint stimulation; transcutaneous spinal electroanalgesia; and 5 KHz sine wave currents.

Previous systematic reviews on TENS had considered some of these techniques as ‘TENS’ despite the electrical characteristics of currents deviating from those defined as standard TENS [630]. For example, Itoh et al. claimed to evaluate TENS for osteoarthritis of the knee [480] and for non-specific low back pain [481] yet devices were delivering interferential currents “… *a single-channel portable TENS unit (model HVF3000, OMRON Healthcare Co Ltd., Japan), which sends between two electrodes a premixed amplitude-modulated frequency of 122 Hz (beat frequency) generated by two medium frequency sinusoidal waves of 4.0 and 4.122 kHz (feed frequency)*” [481] p. 23. Both studies were included in systematic reviews on TENS [636,637].

### 3.5. Inappropriate Body Site

At least 20 studies applied TENS to acupuncture points that were not close to the painful site, commonly using transcutaneous electric acupoint stimulation (TEAS, TAES) and ‘dense-disperse’ currents alternating between 2 pps and 100 pps. Some studies administered transcutaneous electric acupoint stimulation before surgery to manage post-surgical pain. Often details about transcutaneous electric acupoint stimulation treatment protocols were unclear. Four studies were excluded for delivering TENS intravaginal [423,527,528] or intra-oral [519].

### 3.6. Features of Included Studies

The characteristics of the included studies are summarised in the primary report of the Meta-TENS study [12].

### 3.7. Study Design

Studies were conducted in 43 countries with Turkey, the United States, Brazil, the United Kingdom and Sweden being commonest. The majority of studies were parallel-group, pragmatic (clinical) rather than exploratory (mechanistic) and did not estimate sample size a priori (Figure 3).

There were 381 studies, 383 population samples and 24,532 participants. Study sample size was 64.05 ± 58.29 participants (mean ± standard deviation (SD); maximum = 607 [63], minimum = 5 (370)) and TENS group size was 27.67 ± 21.90 participants (*n* = 10,596 participants, maximum = 144 [63]; minimum = 5 participants [26,93,101,370,382]). There were 13 studies with 100 or more participants in the TENS group (Figure 4), yet there was extractable data for only two studies (labour pain [352] and fibromyalgia [96]). There were 341 studies with less than 50 participants in the TENS group.

### 3.8. Types of Pain

There were similar proportions of studies evaluating acute pain and chronic pain, and only a small proportion of studies evaluating both acute and chronic pain (Figure 5).

The commonest pain condition according to nomenclature used by study authors was post operative pain, followed by non-specific musculoskeletal pains and osteoarthritis (Figure 6). Challenges arose categorising pain condition because reports lacked detail about presenting features such as the presence of neuropathic elements, e.g., post-operative pain and post-stroke pain may present as primarily musculoskeletal, neuropathic or both. We created an operational aide memoire to improve the consistency of categorising pain conditions (medical diagnoses).

### 3.9. Treatment Comparators

#### 3.9.1. Most Studies Had No More Than Two Comparator Groups

Approximately two-thirds of studies had one comparison groups and one-third had two comparison groups (Figure 7).

#### 3.9.2. Similar Proportions of Placebo or Active Treatment Comparators

Our definitions for comparison groups were as follows:
Placebo: an inactive intervention that looks the same as, and is given in the same way as, active TENS or active treatment (e.g., drug pill).No treatment: participants did not receive any ‘active treatment’, including background or rescue medication or treatment.Standard of care: intervention(s) that study authors stated to be routine, common or standard care or practice.Other treatment(s): treatment not previously categorised as standard of care (SoC).

There were 476 comparators in 381 studies, 54.2% were an active treatment (i.e., a SoC (26.7%) or other treatment (27.5%)) and 42.4% were a placebo (Figure 8). Thirty-seven studies evaluated low versus high frequency TENS.

The majority of placebo comparators used a TENS device with a dead battery or with modified circuitry so that there was 0 mA current (Figure 9). Other placebo TENS interventions used TENS devices that delivered currents for less than one minute after which current amplitude declined to 0 mA, or long interpulse intervals claimed to be unlikely to produce physiological effects. Some placebo interventions administered TENS below the threshold for sensory detection or to locations unrelated to the painful site.

### 3.10. Many Instances of Contamination from Concurrent Treatment

At least 216 of the 383 study samples were able to use other treatments whilst receiving TENS. Commonly, analgesic medication and/or exercise was combination treatment that was added as part of the study design or ongoing clinical care not considered to be part of study design. Often participants could access other treatments as background or rescue interventions in studies claiming that TENS was delivered as a sole treatment. Some studies monitored and/or standardised rescue medication but monitoring concurrent treatment was inadequate. Thus, unequal contamination from concurrent treatment between intervention groups could not be discounted. Improved monitoring and reporting of concurrent treatment are needed in future studies.

### 3.11. Features of TENS Intervention

#### 3.11.1. Location of TENS and Pain

The majority of TENS interventions were located at the painful site; acupuncture points at the painful site were occasionally chosen [84,152,169,258,363,389,391] (Figure 10). TENS was not administered at the painful site if there was heightened sensitivity and/or poor integrity of the skin associated with painful diabetic neuropathy (TENS administered to the back-dermatomal [66,370]); and for phantom limb pain (TENS administered to the contralateral limb [355]). There were two reports with an unclear statement of TENS location [179,255] and two reports not stating TENS location; in both instances, information within reports confirmed that TENS was delivered at the painful site [171,323].

#### 3.11.2. Intensity of TENS

The majority of reports stated that TENS was delivered above the sensory detection threshold and at a strong non-painful intensity (Figure 11). The intensity of TENS was not reported for 34 samples and unclear for 7 samples, although it was possible to establish that the intensity of TENS was above sensory detection threshold from other information in reports (e.g., reporting of mA or statements in the main text).

#### 3.11.3. Electrical Characteristics of TENS—Pulse Frequency

Most reports provided sufficient detail to confirm that the electrical characteristics of TENS were consistent with that available from a standard TENS device (i.e., pulsed currents, frequency ≤250 pulses per second (pps), pulse width (duration) ≤500 μs and amplitude ≤60 mA (peak-to-peak), Figure 12a). Extracting specific information about electrical characteristics was challenging with 9 reports not stating TENS parameters and 11 reports providing unclear information (other information such as device model was used to confirm that characteristics were consistent with standard TENS.

Terms used to describe TENS technique were inconsistent and included conventional, acupuncture-like, brief intense, high- and low-frequency and acu-TENS. There is inconsistency in units of measure to describe pulse frequency with Hertz (Hz) used most frequently and pulses per second used occasionally. We will use Hz if study authors stated Hz in their study report and pulses per second (pps) when referring to our categorisation of pulse frequency.

We categorised the majority of TENS interventions as high-frequency TENS (>10 pps) and a small minority of 35 samples receiving low-frequency TENS (<10 pps and/or <10 trains of pulses (bursts) per second, Figure 12b). Often reports describing low-frequency stimulation did not differentiate bursts per second from pulses per second making it difficult to establish whether low-frequency pulses or low-frequency bursts of high-frequency pulses were used.

It was less common for reports to state the pattern (mode) of pulse delivery; we presumed high-frequency currents were administered using a continuous pattern of pulse delivery when pulse pattern statements were absent and confirmed this by checking the design of the device. There were 17 samples that received pulse patterns alternating (or switching) between burst and continuous patterns and 9 samples that received alternating (switching) between low and high-frequency pulses (Figure 13a). Ten samples received modulating pulse frequency between upper and lower boundaries, two samples received random pulse frequency and six samples received various pulse frequencies.

Of the 353 reports that stated pulse frequency, 109 stated that 100 Hz was used and 11 stated that TENS was delivered at frequencies below 5 Hz (Figure 13b). A large proportion of reports stated numerous pulse frequencies than included situations where participants received instructions to adjust frequency as needed, or TENS was delivered using modulating or alternating frequencies. It was often unclear whether frequencies were static throughout treatment or whether participants could adjust frequency according to need.

#### 3.11.4. Adequacy of TENS Intervention

There were 336 samples where the report clearly provided details about electrical characteristics, body site of TENS and the intensity of stimulation above sensory detection threshold, and 47 samples where this was not the case.

Regimens included treatments that delivered as a single dose or multiple doses. The shortest duration treatment was a few minutes (e.g., post-partum contraction pain [264], dysmenorrhea [265], surgical abortion [286] or laparoscopic surgery [287], procedural pain (e.g., carboxytherapy) [310]) to as long as needed (e.g., home treatment self-administered prn). One study advised participants to self-administer TENS for chronic pain until they no longer required TENS or until the study ended at 2 years [255].

### 3.12. Features of Outcome Measures

There were 352 studies that collected pain intensity data as the primary outcome and 29 studies that collected pain intensity data as a secondary outcome. The choice of other outcome measures depended on the clinical condition under investigation, e.g., WOMAC would only be selected in studies using population samples with osteoarthritis whereas analgesic consumption could be used for any study population. Analgesic consumption (127 studies) was most common. Other common outcome measures were 52 studies using range of motion, 26 studies using the McGill Pain Questionnaire, 23 studies using blunt pressure pain via algometry, 14 studies using WOMAC, 12 studies using Quality of Life and 8 studies using the Roland Morris Disability Questionnaire.

We explored outcome data collected as the last measurement during TENS or the first measurement post-TENS. There were 91/202 studies (92/203 samples, 4841 participants) with extractable pain intensity (continuous) data and 3 of these studies were crossover studies [96,248,293]. Only two of these samples had 100 participants or more in the primary TENS group (Thomas et al. [352]—labour pain and Dailey et al. [96]—fibromyalgia); and nine samples had at least 50 participants in the primary TENS group with extractable data.

We detected two instances where study reports described outcomes in favour of TENS, yet data analyses suggested otherwise [184,225]. It appeared in both instances that the inconsistencies were transcriptional errors. For the study by Kayman-Kose et al. [184], the inconsistency was for vaginal delivery but the caesarean section data. For the study by Luchesa et al. [225]), crossing checking data with a systematic review [638] established that that TENS data had been ascribed to the placebo group and vice versa.

Of the 127 evaluations of TENS versus a SoC intervention, 71 were versus medication (pharmacological), 40 versus exercise/manual therapy, 3 versus medication combined with exercise/manual therapy and 13 studies were not exercise/physiotherapy or pharmacological or were unclear. There were 61 studies (61 samples, 3155 participants) with extractable data. Of the 131 evaluations with other treatment not categorised as SoC, the commonest comparators were interferential therapy, other electrotherapies and manual therapies. There were 67 studies and 131 samples (3327 participants) that had extractable data, although this included duplicate data from some primary TENS groups.

### 3.13. Features of Risk of Bias Assessment

No studies had low risk of bias (RoB) across all 9 RoB items and only 3/381 studies were assessed as having a low RoB across 8 of the 9 items [95,294,362] (Figure 14). Many study reports were sparce on methodological operational details and were categorised as unclear RoB. Insufficient numbers on TENS participants were a serious problem, with 341 out of 381 studies having a high RoB due to having less than 50 participants in the TENS trial arm. No studies met criteria for low RoB (≥200 participants in the TENS group).

Reasons for high or unclear risk of bias across the nine items were:Reports stated participants were randomised to intervention groups but did not specify whether randomisation was constrained or unconstrained or provided operational details of randomisation (e.g., coin toss, random number sequence generation) or allocation concealment (e.g., sequentially numbered, opaque, sealed envelopes or containers, centrally controlled procedures).Partial or unclear reporting meant that it was not possible to determine with certainty whether all participants completed the study.Only partial descriptions for analyses of outcome measures leading to the possibility of overestimation or underestimation of treatment effects (Selective Reporting (Reporting Bias)).Inadequate descriptions for methods of blinding of participants, personnel or assessor.Inadequate sample sizes.Absence of calculation used to estimate study sample size.

It was notable that only 130 described adequate methods of blinding of assessors (i.e., low risk of detection bias) and only 94 described adequate blinding procedures for participants and 48 for personnel (i.e., low risk of performance bias). Few studies assessed blinding leakage or the plausibility and credibility of interventions, especially whether active and placebo TENS devices were considered to be ‘functioning correctly’ [96,221,282,294]. High quality design, delivery and success of blinding of placebo TENS interventions was observed in studies by Dailey et al. [95,96,294]. There were 129 reports that claimed to have estimated sample size and those that included a calculation often estimated study sample size rather than study group sample size. A summary of individual study RoB judgements for individual studies is provided in Figure 15.

### 3.14. Features of Adverse Events

Generally, reporting of evaluation and reporting of adverse events was inadequate with only 136 out of 381 reports including a statement of adverse events (Figure 16).

Evaluation of adverse events was neglected and when present, methodology and reporting were inadequate. Reports failed to distinguish adverse events associated with study conditions (e.g., resulting from interventions or measurement procedures) or a general worsening of a medical condition (e.g., natural fluctuation of symptoms). Ninety reports acknowledged no adverse events associated with TENS. Forty-six reports acknowledged an adverse event in the TENS group although often it was not possible to determine whether adverse events in the TENS group were directly attributable to TENS. TENS-related adverse events were irritation, soreness or tenderness of the skin that were mild and infrequent. Some reports inappropriately ascribed TENS discomfort or worsening pain as an adverse event, although this could be considered a lack of response to treatment. Only one adverse event was deemed serious “*There was a possible relationship between the treatment and spontaneous abortion. A 36-year-old woman had a spontaneous abortion that occurred 21 days after BTX-A injection and electrical stimulation*.” [318] p. 414.

## 4. Discussion

Our descriptive analysis of studies evaluating the efficacy of TENS provides a visual synopsis of key features of TENS studies. TENS literature is characterised by small, parallel group studies conducted in over 40 countries that evaluate high-frequency TENS at a strong non-painful intensity located at the painful area compared with placebo, analgesic medication(s), exercise, manual therapies and electrophysical agents. There were very few instances of duplicate publications or inconsistencies between data within study reports and that were extracted for meta-analyses.

The main weakness of studies is inadequate sample sizes and the escalating rate of inadequately powered studies on TENS is troubling. Ioannidis raised concern about the proliferation of the publication of useless and meaningless research associated with inadequate sample sizes, “*Most clinical research therefore fails to be useful not because of its findings but because of its design*” [639] p. 1. No studies aligned with criteria for undertaking TENS studies based on previous work by Bennett et al. [640] and Sluka et al. [641], and in doing so, the design and execution of TENS would markedly improve. In addition, study reports should focus much more on the robustness of methods, data and analyses than the physiological and clinical plausibility of findings. In future, authors should make better use of supplemental material to overcome restrictions in manuscript word counts.

### 4.1. Considerations for Future TENS Studies

#### Adequate Sample Size

Inadequate sample size is the most serious shortcoming in TENS studies. Calls for large multicentre studies date back to the 1970s and have been a consistent conclusion of systematic reviews (see Johnson 2021 for review [11]). Readers should be mindful of apparently large study sample sizes that are distributed across multiple intervention groups, seriously compromising statistical power and the likelihood of detecting a true effect [642]. Excessive use of comparison groups and outcome measures complicate interpretation of findings. We discourage the use of multiple intervention groups at the expense of group sample size in future studies.

### 4.2. Reframing Blinding of TENS

Performance bias is central to debates about the quality of studies on TENS. Strategies to judge blinding of participants and therapists have varied in previous reviews with some reviewers arguing that TENS studies always have a high RoB because blinding participants to TENS sensations is not possible as participants will know whether they are receiving an active or placebo intervention. We counter-argue that whether participants are uncertain about the intervention ‘functioning properly’ is the crucial factor, as this creates the belief that a placebo intervention may be a credible and therefore potentially active treatment. Pre-study briefings can be used to generate uncertainty in participants and personnel about whether a device needs to generate a sensation to be functioning properly and a TENS device that has been adapted so that there is no current output can be used [643,644]. This was the case for many of the studies included in this review. However, at present there is a paucity of studies that evaluate the outcome of blinding.

### 4.3. Managing Contamination from Concurrent Treatment

There were many instances of concurrent treatment and inadequate monitoring and/or reporting of concurrent treatment. Contamination from concurrent treatment is an issue in clinical trials [645]. In pain studies, differential adjustment of concurrent treatment (analgesics) between groups may generate comparable pain intensity outcomes [646] and undervalue the extent of beneficial effects [640]. Improved monitoring and reporting of concurrent treatment are needed in future studies.

### 4.4. Evaluating Adverse Events

Most TENS studies focus on benefits, and it was rare for adverse effects to be pre-specified as an outcome. Commonly, adverse event data were captured by ad hoc observation resulting in a high risk of selective reporting bias. This is not unique to TENS [647]. In future, methodologies to obtain data for adverse events should be formalised with clear criteria to ascribe the occurrence and seriousness of adverse events.

### 4.5. Reframing Outcomes

There were similar proportions of studies evaluating acute pain and chronic pain; the commonest evaluations were for post-operative pain with a paucity of studies with extractable data for prevalent chronic pain conditions, including non-specific neck and/or back pain, osteoarthritis and neuropathic pain. At present, there is inconsistency in the National Institute for Health and Care Excellence (NICE) guidelines in the United Kingdom where TENS is recommended as an addition to primary treatment for osteoarthritis [648] and rheumatoid arthritis [649], but not recommended for non-specific chronic low back pain [650], chronic pain [651] and intrapartum care (labour pain) [652].

Our Meta-TENS study found moderate-certainty evidence of pain relief *irrespective of diagnosis* [12].

We argued that pathological indicators that underpin medical diagnoses (pain conditions) may be of little relevance for treatments that alleviate pain using soothing sensations (e.g., TENS, warmth, cooling, touch). This is because the complex nature of the lived experience of pain results from much more than pathological indicators that may or may not be generating nociceptive input—everything matters for pain. Pain is a warning signal to protect tissue and is not always a dependable monitor of tissue damage (status), i.e., hurt does not always mean harm. Thus, we suggest that it may be more appropriate to select such treatments based on the quality of pain sensation.

Our descriptive analysis found few instances of studies allowing participants to select outcomes that were important to them. In clinical practice, TENS is selected following a holistic evaluation of a person’s lived experience of pain, irrespective of medical diagnosis, and in line with a biopsychosocial self-management framework. Long-term users of TENS report using it to relieve sensations of pain and muscle spasm, thus reducing the negative impact of an ‘overprotective brain’ by enhancing function, sleep, psychological well-being and medication reduction [653,654,655]. TENS can promote activities of daily living and quality of life when used in combination with pain education, lifestyle adjustments, and movement and psychological-based interventions. Thus, outcome measures should document immediate relief of pain and function ‘in-the-moment’ during TENS, and longer-term effects over a prolonged course of TENS treatment. Careful consideration should be given to techniques used to measure follow-up effects after a course of treatment because, for example, a participant may stop using TENS because pain has resolved or because of lack of benefit or intolerable adverse events.

Research supports advice to personalise TENS treatment to personal need using systematic trial and error to find electrical characteristics and electrode positions that maximise benefit and minimise problems [654]. Gladwell et al. argue that future TENS studies need to be designed to align potential benefits of using TENS with patient reported outcome measures (PROMS) using the International Classification of Functioning, Disability and Health (ICF) [656]. This should be underpinned by foundational research to improve evaluations of TENS [657].

### 4.6. Reframing the Active Ingredient of TENS

The TENS study literature is contaminated by TENS-like interventions using non-standard currents or techniques including interferential therapy, transcutaneous electrical acupoint stimulation or microcurrent and therefore it is important to distinguish TENS from TENS-like interventions including administering TENS to acupuncture points remote to the site of pain. The eligibility criteria for our review were optimised to improve the likelihood of beneficial effects from TENS and were based on a pragmatic assumption that the effects of TENS are optimal when a standard TENS device administers strong non-painful TENS sensation close to the painful site. This is consistent with the physiological rationale of TENS to selectively stimulate non-noxious low threshold cutaneous peripheral afferents because this has been shown to reduce activity in nociceptive transmission cells in the central nervous system [2,3,4,5,658]. Studies using animal models of nociceptive processing suggest that central neuropharmacological actions of TENS are influenced by the pulse current frequency [659], yet this may not translate into clinically predictable outcomes in humans [660]. We argue that searching for optimal TENS parameters for specific pain conditions should be abandoned [11].

Our descriptive analysis revealed that the vast majority of TENS interventions were administered using high-frequency currents (>10 pps). However, reporting on electrical characteristics of TENS used in studies needs to be improved. Many reports did not state the pattern (mode) of pulse delivery, which is necessary to differentiate bursts per second and pulses per second when referring to low-frequency TENS. Likewise, the terms ‘alternating’, ‘varying’ and ‘modulating’ were not accompanied with sufficient detail to ascertain the precise nature of currents such as frequencies that switch between upper and lower boundaries at a single time point (i.e., 6 s at 10 pps and 6 s at 100 pps) or over a period of time (e.g., 6 s linear increase in frequency from 10 pps to 100 pps).

In most instances, TENS settings were determined by the researcher with insufficient information to ascertain whether settings could subsequently be adjusted by participants according to need. In most instances, the variability in treatment regimens reflected clinical context, e.g., a single five-minute dose during procedural pains to regular 30–60 min doses as needed over a period of weeks. This level of detail is important to ascertain exactly how TENS was administered throughout a study and whether participants adhered to clinical practice guidelines.

Much debate has focused around optimal and appropriate electrical characteristics of TENS. We have argued that differential neurophysiological and pharmacological pulse frequency effects of TENS observed in animal models of nociception may not moderate clinical outcome, i.e., frequency-dependent physiological effects do not translate to clinically meaningful outcomes. We speculate that the critical factor for success of TENS is the ‘comfortability’ of the sensory experience of TENS as this will ‘soothe’ the intensity and quality of and reduce perceived bodily threat, akin to rubbing, warming or cooling the skin for pain relief. One advantage of TENS is that electrical characteristics can be adjusted to alter the quality of sensations such as pulsate and paraesthesiae, and such moment-to-moment adjustment may be beneficial to combat the dynamic nature of pain.

It may be prudent to consider the ‘active ingredient’ of TENS as a pleasant bodily-TENS sensation irrespective of the electrical characteristics needed to achieve this. It is likely that electrical characteristics of TENS need to be regularly adjusted within and between treatment sessions to maintain this sensory comfortability on a moment-to-moment basis. Studies exploring the lived experiences of successful TENS users are few [653,654] and more are needed.

### 4.7. Monitoring Adherence and TENS Usage Patterns

Most study measurements obtained for in-patient populations reflected real-world situations where TENS was administered and outcome was measured during routine care, e.g., for procedural or post-surgical pain. However, this was apparent for out-patient populations self-administering TENS at home. Often participants were required to attend a ‘study visit’ to the clinic or laboratory to measure TENS effects at that one instance in time, occasionally supported with pain diary data. There was a paucity of formal monitoring of the pattern of TENS usage or whether participants had adhered to instructions for use in out-patient and in-patient settings. The use of sophisticated TENS devices that record patterns of treatment should be encouraged in future studies. Likewise, there is an urgent need for TENS education packages to support the intricate behaviours and choices facing TENS users so that they can optimise TENS benefits both in clinical practice and research studies [657].

### 4.8. Consideration of Novel Study Designs

The majority of TENS studies were parallel group randomised controlled trials conducted in hospital settings and using short-term outcomes. Two-thirds of the studies were pragmatic (clinical) rather than explanatory (mechanistic) with a large proportion of evaluations for post-operative pain and chronic musculoskeletal pain (e.g., back pain and osteoarthritis). The paucity of large studies with long-term courses of treatment and long-term follow-up for common chronic pain conditions should be addressed in future research.

There were no enriched enrolment randomised withdrawal studies. Previously, we have argued the need for a large multicentred enriched enrolment randomised withdrawal study that includes an observational ‘run-in’ phase of at least two weeks to enable participants to tailor treatment to their needs (i.e., optimise treatment and troubleshoot problems) and facilitate the assessment of adverse effects [11]. Group sample sizes should be 200 participants or more to ensure adequate statistical power [11,642], although we suspect the effect size estimate of such a study would be of a similar magnitude to that reported in our Meta-TENS study (i.e., pain intensity was lower for TENS versus placebo with a standardised mean difference of −0.96 (95% CI −1.14 to −0.78; 91 RCTs, 92 samples, *n* = 4841 [12]).

There is a need for real-world data so that educational packages can be developed to support patients to self-administer TENS and to integrate TENS within public health service settings [655,657]. Research is also needed to determine appropriate, contextualised TENS patient-reported outcomes [656].

## 5. Conclusions

The analyses offer insights into the factors influencing the fidelity of studies on TENS and offers avenues to improve the direction and design of future research. The conduct and publication of small-sized studies claiming to evaluate efficacy should be discouraged, as should new systematic reviews without the inclusion of large studies that are likely to change outcome or improve the level of certainty of evidence. Our descriptive analysis should be considered alongside the moderate-certainty evidence that TENS alleviates ‘pain-in-the-moment’ from our Meta-TENS study [12].

## Figures and Tables

**Figure 1 medicina-58-00803-f001:**
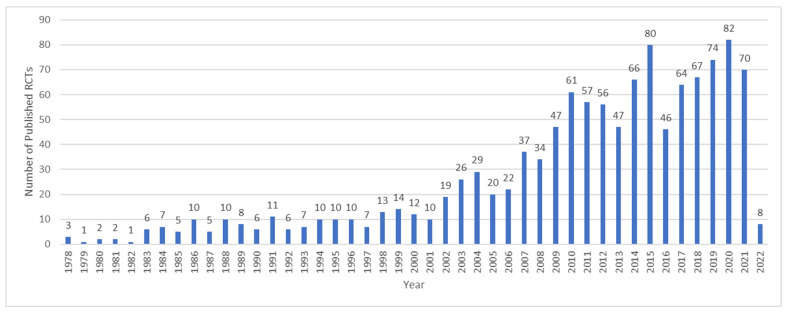
Number of records identified by searching PubMed.gov for randomised controlled trials (RCTs) of TENS and pain: conducted on 31 March 2022. Search String: (transcutaneous electrical nerve stimulation) AND (pain) Filters: Randomized Controlled Trial (“transcutaneous electric nerve stimulation” (MeSH Terms) OR (“transcutaneous” (All Fields) AND “electric” (All Fields) AND “nerve” (All Fields) AND “stimulation” (All Fields)) OR “transcutaneous electric nerve stimulation” (All Fields) OR (“transcutaneous” (All Fields) AND “electrical” (All Fields) AND “nerve” (All Fields) AND “stimulation” (All Fields)) OR “transcutaneous electrical nerve stimulation” (All Fields) AND (“pain” (MeSH Terms) OR “pain” (All Fields))) AND (randomizedcontrolledtrial (Filter)). Note: These records have not been screened against eligibility criteria and will over-estimate the actual number of RCTs.

**Figure 2 medicina-58-00803-f002:**
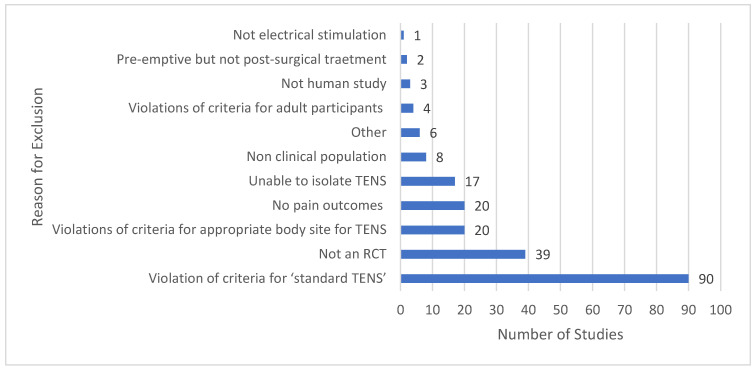
Summary of primary reason for excluding studies: RCT = randomised controlled trial.

**Figure 3 medicina-58-00803-f003:**
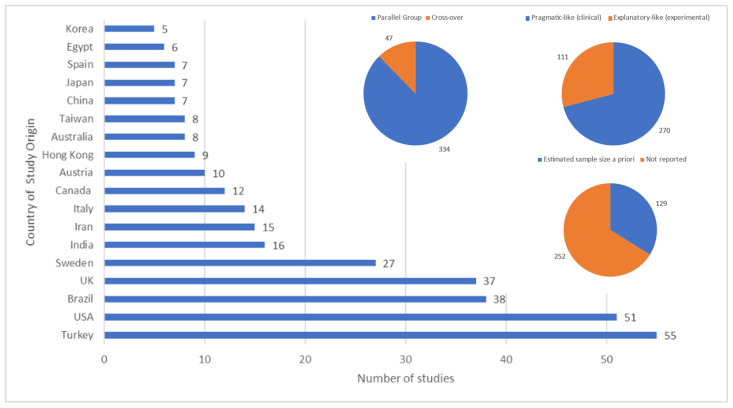
Country of origin of the study; Parallel group or cross-over study design; Tending toward pragmatic or explanatory in focus; Report contained statement that a sample size had been estimated a priori.

**Figure 4 medicina-58-00803-f004:**
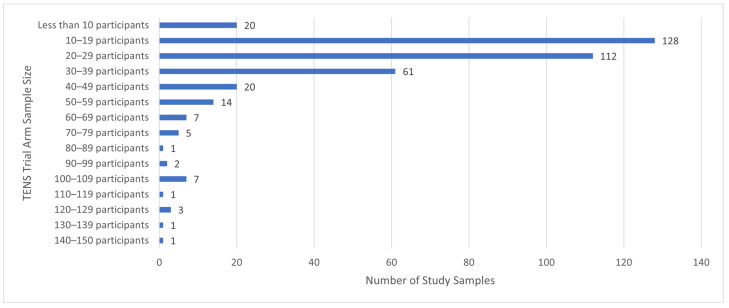
Summary of study sample sizes.

**Figure 5 medicina-58-00803-f005:**
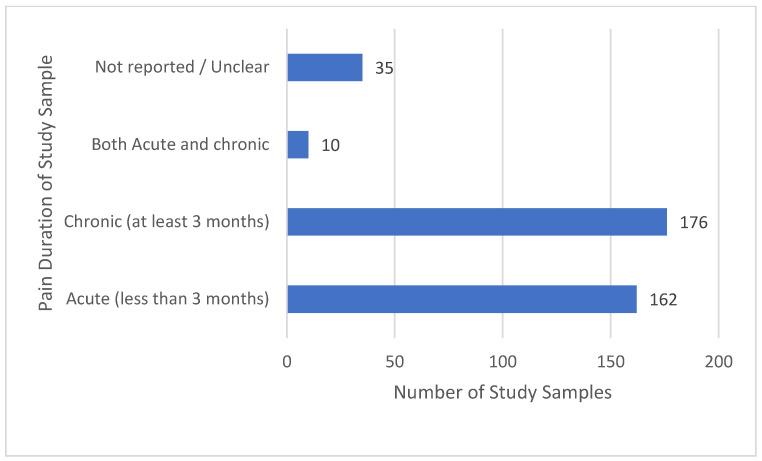
Pain duration.

**Figure 6 medicina-58-00803-f006:**
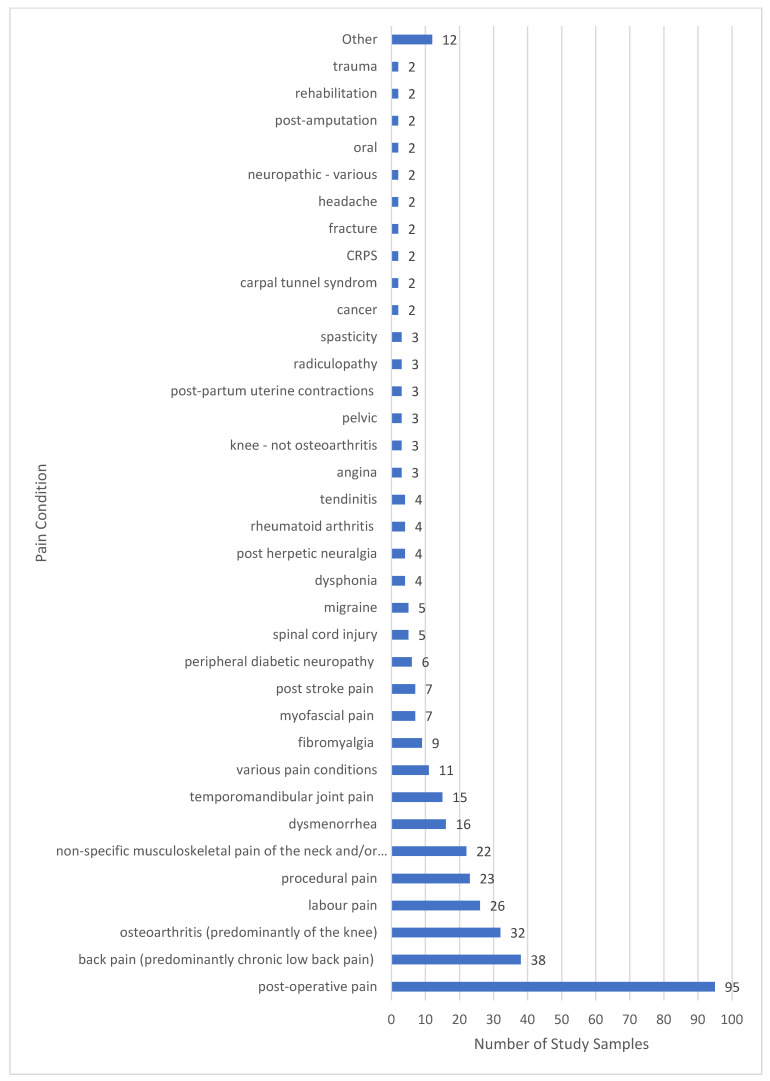
Pain diagnosis categorised as stated by the study authors. Others represents *n* = 1 sample for each of the following: adhesive capsulitis, intercostobrachial pain, plantar fasciitis, haemophilia, ischemia, orchialgia, orofacial pain, pancreatitis, chronic breast cancer treatment pain, pressure ulcers, renal colic and trigeminal neuralgia.

**Figure 7 medicina-58-00803-f007:**
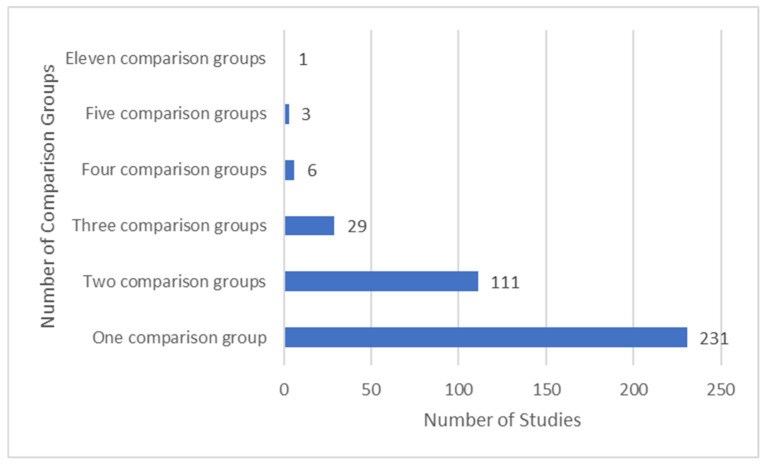
Summary of number of comparator groups.

**Figure 8 medicina-58-00803-f008:**
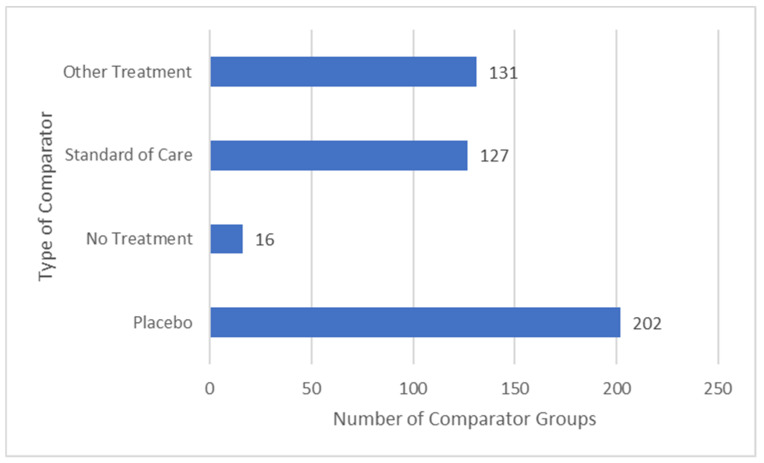
Types of comparator groups.

**Figure 9 medicina-58-00803-f009:**
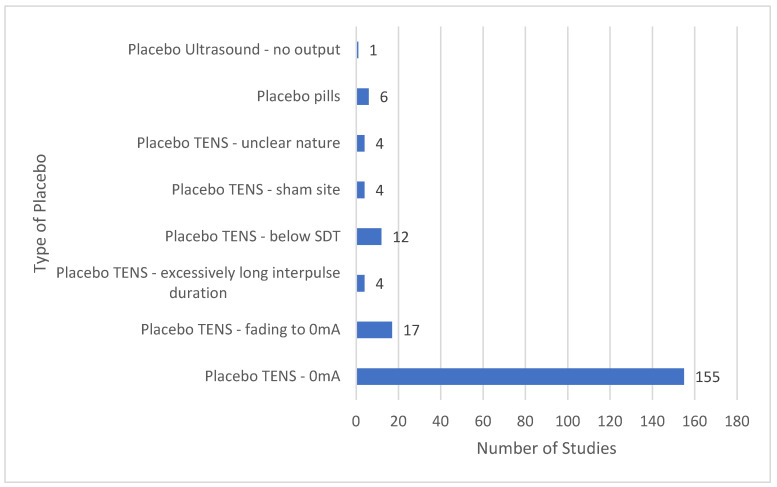
Types of placebo comparators: SDT = sensory detection threshold; sham site = sites unrelated to pain.

**Figure 10 medicina-58-00803-f010:**
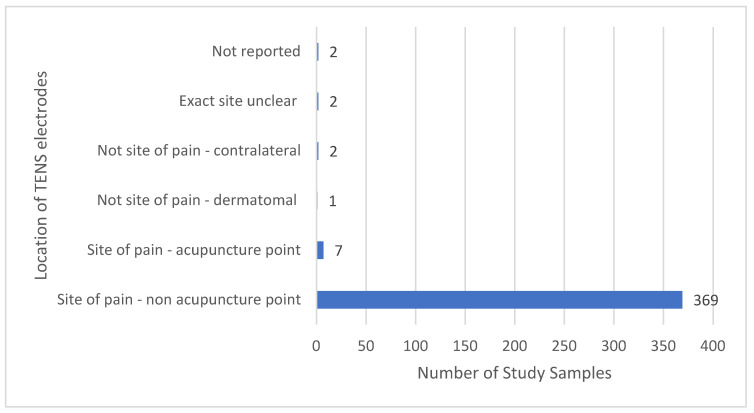
Location of TENS relative to pain.

**Figure 11 medicina-58-00803-f011:**
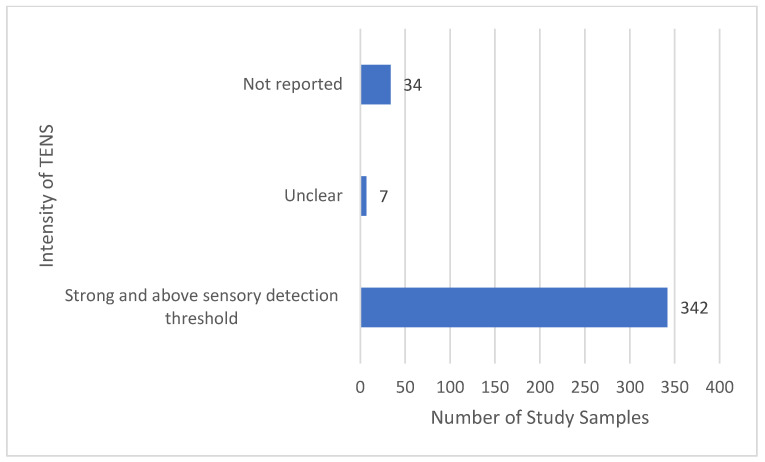
Intensity of TENS.

**Figure 12 medicina-58-00803-f012:**
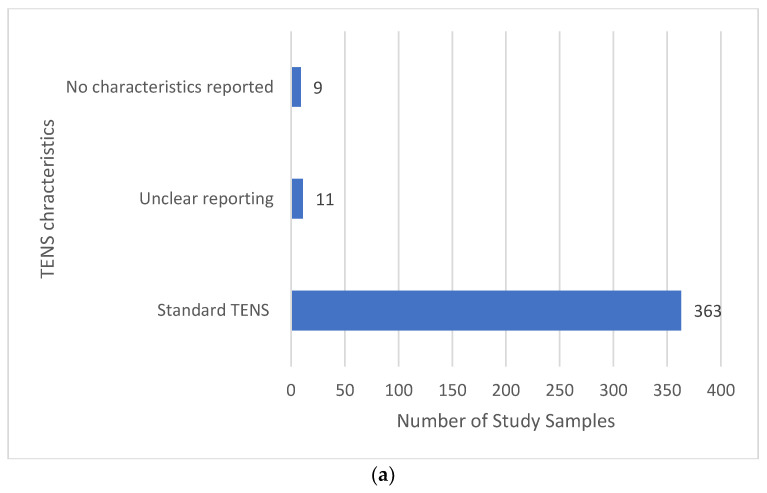
(**a**) Electrical characteristics of TENS; (**b**) type of TENS: * Using a continuous pulse pattern. ** Either unclear or prn.

**Figure 13 medicina-58-00803-f013:**
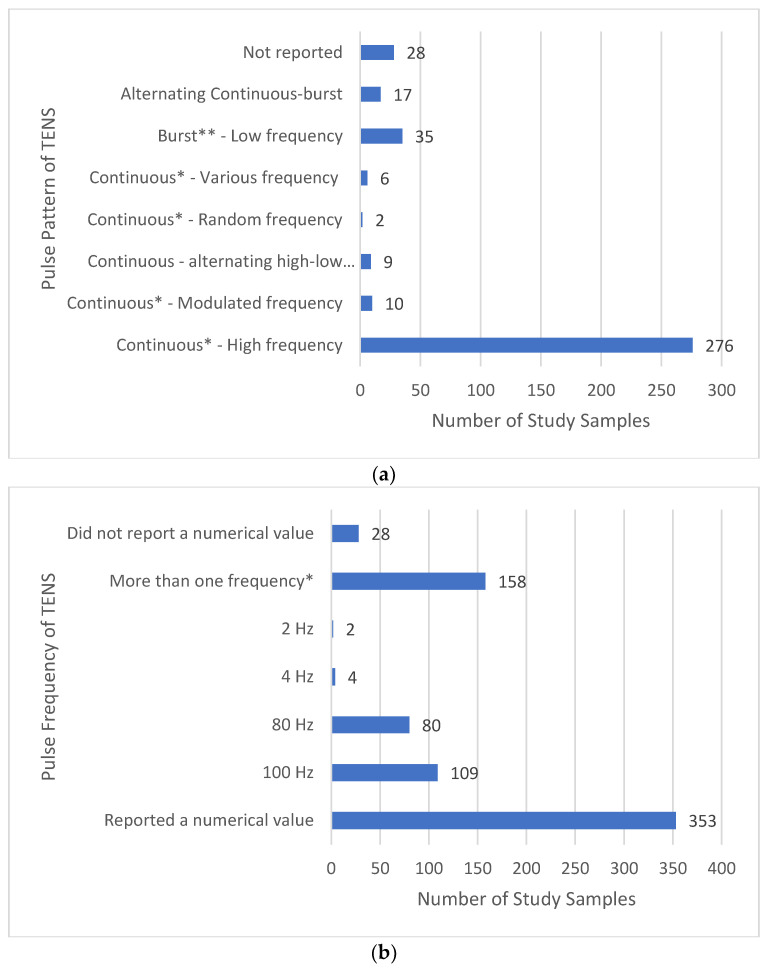
(**a**) Pattern (mode) of pulse delivery of TENS: * We were able to infer that a continuous pattern of pulse delivery was used to deliver high frequency currents in instances where the pattern of pulse delivery was not clearly stated in the study report. ** We were able to infer that a low frequency bursts (trains) of high frequency pulses were used in instances where the pattern of pulse delivery was not clearly stated in the study report; (**b**) frequency of pulse delivery of TENS: * Including instances where modulated or alterating pulse frequencies were used or frequencies were prn.

**Figure 14 medicina-58-00803-f014:**
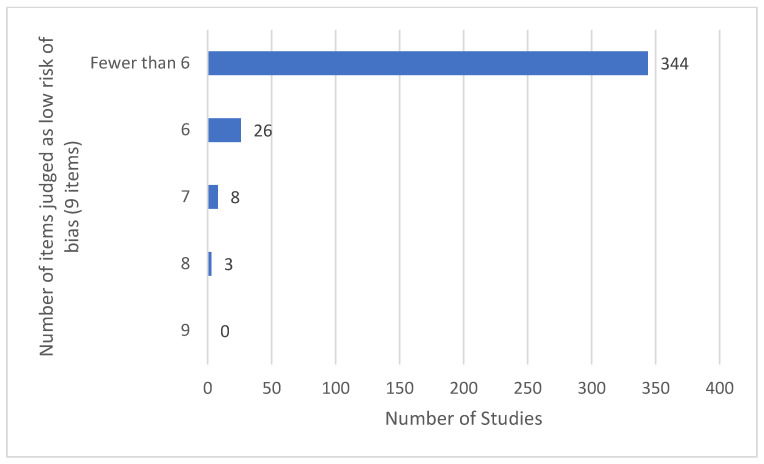
Scores of overall low risk of bias for the 381 studies.

**Figure 15 medicina-58-00803-f015:**
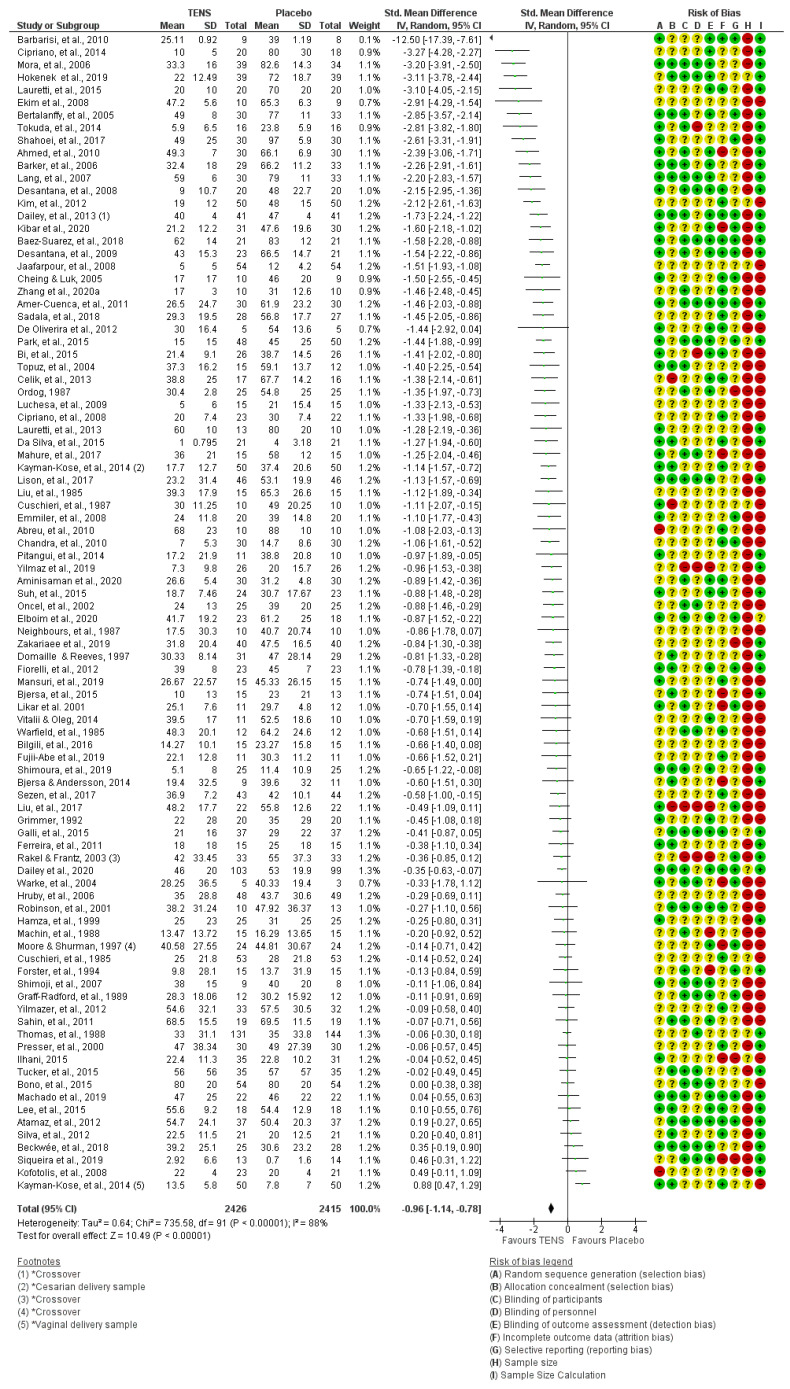
TENS versus placebo for pain intensity (continuous data) as standardised mean difference (SMD), including individual risk of bias (RoB) judgements. Green + circles = low RoB, Yellow ? circles = unclear RoB, Red circles = high RoB, * = calls attention to the footnote: ◂ = SMD and 95% confidence intervals lie outside the range of the horizontal axis −12.5 (−17.39, −7.61): ♦ = Overall measure of effect, the lateral points indicate confidence intervals for this estimate. The sequence of the in-figure reference citations from top to bottom are [43], [86], [249], [166], [201], [115], [52], [356], [321], [19], [44], [198], [104], [187], [95], [186], [39], [105], [176], [75], [393], [30], [310], [101], [277], [53], [358], [71], [268], [225], [85], [200], [94], [230], [184], [221], [222], [92], [118], [16], [73], [285], [386], [31], [346], [266], [116], [258], [392], [112], [135], [237], [57], [213], [377], [381], [54], [140], [325], [56], [320], [223], [152], [141], [132], [293], [96], [382], [168], [305], [156], [229], [248], [91], [139], [324], [149], [387], [311], [352], [289], [173], [364], [60], [228], [206], [36], [329], [48], [332], [192] and [184].

**Figure 16 medicina-58-00803-f016:**
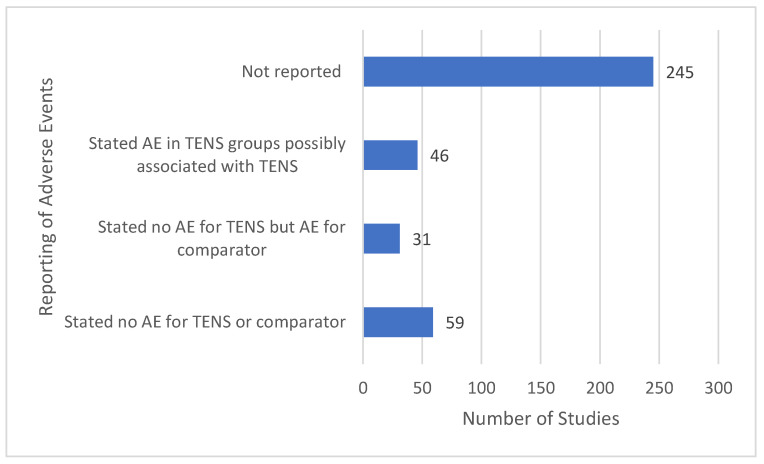
Tally of adverse event statements.

## Data Availability

The data presented in this study are available on request from the corresponding author. The data are not publicly available as yet due to the investigating team continuing to undertake secondary analyses.

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
