# Peer review of "Characterising the Features of 381 Clinical Studies Evaluating Transcutaneous Electrical Nerve Stimulation (TENS) for Pain Relief: A Secondary Analysis of the Meta-TENS Study to Improve Future Research"

_medicina, 2022, doi:10.3390/medicina58060803_

Round 1
Reviewer 1 Report
This work is both important and interesting in the way that it details trends in TENS research. It has strong potential to improve the design of future TENS research. I found only minor issues with grammar and presentation which I think can be simply resolved.
On page 7, there is a typo: "p[resented". Should this be "represented"?
On page 11, I think there is an extra word (if) in this sentence: "...painful site if in..."
On page 14, I think there is a duplication of "minute": "few minutes minute".
On page 15, please check the syntax, there may be a missing word: "We detected data two instances"
On page 15, in the bottom paragraph, there is an unnecessary space or two after RoB.
Figure 15: this was hard to read due the small text size, I wondered if it could be split over two pages? This would help the reader, especially if the article is printed out on paper, and the figure descriptor would also fit on the same page.
Page 19: "participants and therapists has" should I think be "participants and therapists have"
Page 19: the tense of "undervaluing" might be better here as "undervalue".
Page 20: "foundational to improve" reads as if there might be a missing word, such as "foundational research to improve"
Page 22: I though that it might help the reader if the magnitude of the effect size was stated clearly here: "we suspect the effect size estimate of such a study would be of a similar magnitude to that reported in our Meta-TENS study."
Page 22: should "pain-in-the-moment" have quotation marks at either end? It only has one at the end at present.
Author Response
Thank you for taking the time to review our manuscript and for your comments and suggestions. We agree with all of your suggestions and have amended the manuscript accordingly.
On page 7, there is a typo: "p[resented". Should this be "represented"?
- amended
On page 11, I think there is an extra word (if) in this sentence: "...painful site if in..."
- amended
On page 14, I think there is a duplication of "minute": "few minutes minute".
- amended
On page 15, please check the syntax, there may be a missing word: "We detected data two instances"
- amended
On page 15, in the bottom paragraph, there is an unnecessary space or two after RoB.
- amended
Figure 15: this was hard to read due the small text size, I wondered if it could be split over two pages? This would help the reader, especially if the article is printed out on paper, and the figure descriptor would also fit on the same page.
- I have provided an original JPG of the figure and have asked the proof editor to advise on how best to present this figure and caption in the final manuscript version
Page 19: "participants and therapists has" should I think be "participants and therapists have"
- amended
Page 19: the tense of "undervaluing" might be better here as "undervalue".
- amended
Page 20: "foundational to improve" reads as if there might be a missing word, such as "foundational research to improve"
- amended
Page 22: I though that it might help the reader if the magnitude of the effect size was stated clearly here: "we suspect the effect size estimate of such a study would be of a similar magnitude to that reported in our Meta-TENS study."
- added
Page 22: should "pain-in-the-moment" have quotation marks at either end? It only has one at the end at present.
- amended
Kindest regards
Reviewer 2 Report
Dear Authors,
I've read your work with pleasure. It is a continuation of a comprehensive review of data obtained for huge group of studies.
I have few minor comments:
- please correct citations' numbering (e.g. Bennett and further)
- do you have additional data regarding quality of life assessment in these studies? If you do that would be interesting to read separate paragraph.
- there is huge number of records (e.g. in PubMed) when we serch for TENS, cancer and pain. Could you explore or comment? Looking at Figure 6 it looks like only 3 studies were related to cancer pain (2+1).
There is quite a number of self citations, but looking at the fact that authors have been publishing for quite a long time in that field and the number of citations in general that is acceptable.
The article is within scope of the special issue and I am happy to accept it with some minor comments.
Regards,
Reviewer
Author Response
Thank you for your careful consideration of our manuscript. We have amended the manuscript in agreement with all of your suggestions.
I have few minor comments:
- please correct citations' numbering (e.g. Bennett and further)
- Thank you ever so much for detecting this. Managing such a large volume of citations using the Vancouver system is complex even when using Endnote citation management software. We have detected the original errors in numbering and have amended the reference list and in-text citations
- do you have additional data regarding quality of life assessment in these studies? If you do that would be interesting to read separate paragraph.
- Unfortunately, we have as yet been unable to conduct any substantive analyses of non-pain outcomes such as quality of life. It is something that we hope to do in the future.
- there is huge number of records (e.g. in PubMed) when we serch for TENS, cancer and pain. Could you explore or comment? Looking at Figure 6 it looks like only 3 studies were related to cancer pain (2+1).
- We agree that a search of PubMed for TENS, cancer and pain would generate many records. However, the majority of these 'hits' did not meet the eligibility criteria for our review (the Meta-TENS study) mostly because they were not RCTs. In fact there were only 3 RCTs after screening and this number is consistent with the most recent Cochrane review on the topic by Hurlow et al.
There is quite a number of self citations, but looking at the fact that authors have been publishing for quite a long time in that field and the number of citations in general that is acceptable.
- Yes, we absolutely agree. We were extremely mindful of the issue of self-citations when preparing the manuscript. We believed that some self-citations were necessary to convey the historical context in which our programme research developed over time. We are pleased that you believe the extent of self-citation is acceptable.
Kindest regards and thank you for taking the time to review our manuscript
This manuscript is a resubmission of an earlier submission. The following is a list of the peer review reports and author responses from that submission.